# Gold Mine Wooden Artefacts: Multianalytical Investigations for the Selection of Appropriate Consolidation Treatments

**DOI:** 10.3390/molecules27165228

**Published:** 2022-08-16

**Authors:** Mariusz Fejfer, Jeannette Jacqueline Łucejko, Beata Miazga, Emma Cantisani, Magdalena Zborowska

**Affiliations:** 1Archaeological Museum in Biskupin, Biskupin 17, 88-410 Gąsawa, Poland; 2Department of Chemistry and Industrial Chemistry, University of Pisa, Via G. Moruzzi 13, 56124 Pisa, Italy; 3Institute of Archaeology, University of Wrocław, Szewska 48, 50-139 Wrocław, Poland; 4Institute of Heritage Science—National Research Council of Italy, Via Madonna del Piano 10, Sesto Fiorentino, 50019 Florence, Italy; 5Department of Chemical Wood Technology, Faculty of Forestry and Wood Technology, Poznań University of Life Sciences, Wojska Polskiego 38/42, 60-637 Poznań, Poland

**Keywords:** waterlogged wood, minerals, Py-GC/MS, FT-IR, ED-XRF, SEM-EDS

## Abstract

Environmental conditions present in mines generally are very favourable to decay; high temperature, high humidity, variable oxygen content, numerous metal-wood connections and the presence of a high content of inorganic compounds typical of mines have a significant impact on the biotic and abiotic degradation factors. The state of conservation of wooden artefacts from the Złoty Stok (Poland) gold mine was investigated using a multi-analytical approach. The aim was to select the conservation treatments that would stop decay and improve the conditions and dimensional stability of the wood. FT-IR and Py-GC/MS were used to assess the state of preservation of lignocellulosic material. ED-XRF and SEM-EDS were used to determine—and XRD to identify crystalline phases—salts and minerals in the wood structure or efflorescence on the surface. Highly degraded lignocellulosic material that had undergone depolymerisation and oxidation was found to be severely contaminated by iron-based mineral substances, mainly pyrite, and in some cases greigite and magnetite. The presence of inorganic salts made it difficult to choose the best consolidating material to reduce the level of decay and improve the dimensional stability of the wood.

## 1. Introduction

Since ancient times, human activity has been strongly linked to the wood, which is one of the most important natural and sustainable raw materials. Humans have always been surrounded by wood, from simple tools, firewood and furniture, to more complex structures, such as houses, bridges, work machines or musical instruments, works of art and religious objects. It is one of the most resistant organic materials; however, wooden objects can be preserved for long periods of time only under particular conditions. In a temperate climate, these are the anaerobic conditions of wet environments and the aerobic conditions of dry environments, considering the low intensity of wood degradation factors [1,2,3].

Wood has always been and is still used in mines, where specific conditions need to be considered which do not apply to other industrial facilities. Unfortunately, the environmental conditions in mines promote decay. The high temperatures, variable and generally high humidity, variable oxygen content, numerous metal-wood contact points, and the high content of inorganic compounds have a significant impact on the biotic and abiotic degradation factors. In fact, temperature, air supply and a wet atmosphere lead to fungal growth [4], while a high concentration of inorganic salts causes extensive degradation, which results in a severe defibration of the wood [5,6].

In the literature, there have been many studies carried out on wood from various types of mines. Today, in order to prevent mine accidents, the timber framing has to be regularly inspected for decay [4]. Studies on the wooden structures in the mines of the past are quite common, such as salt mines from the Bronze Age [7,8], the mines from the 17th–20th centuries [9,10] and silver, copper and iron mines from the 1st and 2nd centuries [11]. These studies show that the conditions of the wooden structures inside the mines can be greatly compromised. Chemical changes caused by the high salt content in the wood fractions at the polymeric and molecular levels have been determined [9,12]. It is known that some elements can inhibit microbial growth, thus reducing the decomposition of the wood, but at the same time, others can increase their growth. In fact, Wang et al. [13] ranked metals in relation to their toxicity to microorganisms as follows: Cr > Pb > As > Co > Zn > Cd > Cu, while Wyszkowska et al. [14] noted a significant increase in the fungi counts in soil contaminated with Zn and Cu at a dose of 500 mg/kg of soil.

The multiplicity of degradation factors of the archaeological wood from the environment of the mines makes investigations of these artefacts very complex. Furthermore, assessing the state of deterioration of these artefacts for conservation purposes is challenging, precisely because many of these factors need to be considered. Some conservation treatments interact with inorganic salts or metallic elements, causing further deterioration of the wood, such as polyethylene glycol (PEG). In the presence of iron and sulfur, PEG leads to the formation of sulfuric acid, which causes further degradation in the wood [15,16,17].

A wide range of analytical methods can be applied to evaluate the state of preservation of archaeological wood and to determine the causes and processes of decay that have occurred. Infrared spectroscopy (FTIR) is frequently used to assess the degree of wood decay [18] because it is fast and sensitive and when used in attenuated total reflection mode (ATR), it is also non-destructive. It can identify functional groups, differentiate between hardwoods and softwoods and assess the extent of degradation of wood semi-quantitatively. Changes in the chemical composition can be determined by observing characteristic bands for each wood component (hemicelluloses, cellulose and lignin), measuring the heights and the areas of the selected bands and considering ratios between these heights or areas, which can be correlated to the changes occurring [19]. FTIR was used in the study of wood objects on the hyper-saline Dead Sea shore [12].

Another instrumental technique to monitor changes in the chemical composition of wood is analytical pyrolysis (Py-GC/MS). This technique can be used to analyse complex matrices, such as wood, as it allows the selective bond cleavage of wooden polymers, cellulose, hemicelluloses and lignin, and provides compositional information at a molecular level. Py-GC/MS highlights chemical modifications, such as the depletion and depolymerisation of polysaccharides and side chain shortening, oxidation or demethylation occurring in the lignin polymer [20,21,22].

Archaeological wood, as well as other artefacts, is often contaminated by many different substances. These deposits, especially mineral, can be studied by non-destructive and less invasive tools. For this reason, energy-dispersive X-ray fluorescence (ED-XRF) and scanning electron microscopy, coupled with energy-dispersive X-ray spectroscopy (SEM-EDS) are the most common techniques used in the cultural heritage field [23,24,25,26]. XRF provides quantitative information on the elements present in the analysed sample, while SEM-EDS has been used to obtain information on the localisation of inorganic products.

X-ray diffractometry (XRD) is the most commonly used technique worldwide for the identification and characterisation of polycrystalline phases in analysed samples and is a well-established method for determining the crystallinity of partially crystalline materials. Salts and minerals in the wood structure or efflorescence on the surface can thus be investigated, using XRD without any sample pre-treatment [27,28,29,30]. X-ray diffraction is usually used for the calculation of crystallinity, identified as the weight fraction of crystalline material—crystalline cellulose—in wood [31].

In this study, the chemical transformation of lignocellulosic polymers and contamination with inorganic salts in the archaeological wood from the Złoty Stok gold mine were investigated using complementary instrumental techniques: FT-IR, Py-GC/MS, ED-XRF, SEM-EDS 104 and XRD. Due to the unique burial place of the studied elements, it was expected that they would have unusual chemical properties, necessitating the use of special conservation procedures. The aim of the study was to identify the chemical properties of wood, which will be helpful in choosing a preservation method that gives a long-lasting effect of wood stabilisation and protection against further degradation.

## 2. Results

### 2.1. FTIR

The FTIR test was carried out to assess changes in the chemical composition of archaeological wood samples from a gold mine, described in Section 4.1. The results obtained were compared with modern and undegraded pine and spruce reference woods. The spectra obtained were compared in the range of an 1800–800 cm^−1^ fingerprint of wood. The spectra obtained for the archaeological wood (Figure 1 and Figure 2) showed most of the characteristic absorption bands observed in the reference samples. However, in all archaeological wood samples studied, there was no band at 1735 cm^−1^ associated with carboxylic and acetyl groups in the xylans [32]. This phenomenon is very common for archaeological waterlogged wood [12,33,34], but also for modern woods subjected to photo-oxidation, water action or high temperature [18,35].

In all the waterlogged wood samples, a slight increase in absorbance for the selected guaiacyl lignin-related bands (softwood) and a decrease in absorbance for the cellulose and hemicelluloses-related bands were observed. The spectra of waterlogged wood were particularly characterised by an increase in absorbance at 1600 cm^−1^, the band associated with skeletal vibrations in the aromatic ring and the band 1220–1230 cm^−1^ characteristic of vibrations of the C-C, C-O and C=O bonds in the guaiacyl ring of lignin [36,37]. In addition, the pine wood sample P1 showed a significant increase in absorbance at 1267 cm^−1^, related to C=O stretching vibrations and guaiacyl ring vibrations.

The spectra of waterlogged wood of both species were characterised by a decrease in absorbance at 1374, 1158, 1112 and 1057 cm^−1^, bands associated with wood polysaccharide. Hemicelluloses is characterised by a band at 1374 cm^−1^ attributed to C-H bending [36], and a band at 1158 cm^−1^ formed as a result of C-O-C asymmetric valence vibration [38] bonds in hemicelluloses. The 1112 cm^−1^ band is related to the stretching vibration of the C-C and C-O bonds of different groups in holocellulose [38]. The decrease in absorbance at 1057 cm^−1^ related to the C-O bond [38] may be caused by the decay of cellulose and polysaccharide decomposition.

To better highlight differences in the intensities of absorption bands, some semi-quantitative calculations were performed [19], which enable the degradation of wood components to be estimated—especially carbohydrates. Ratios between the relative intensities of lignin peaks at 1510 cm*^−^*^1^ against carbohydrate peaks at 1374, 1158 and 897 cm*^−^*^1^ were calculated, using peak areas and heights [19,37,38]. The peak at 1510 cm^−1^ was chosen as a lignin reference resulting from the skeletal vibration (C=C) of the aromatic ring in the lignin (L). All carbohydrate peaks used for calculations have no significant contribution from lignin, and they are attributed as described previously to C-H deformation in cellulose 897 cm^−1^, and to C–H deformation at 1374 cm^−1^ and to C–O–C vibration at 1158 cm^−1^ in cellulose and hemicelluloses [36].

The results are listed in Table 1. In all archaeological wood samples, the ratios between lignin and holocellulose (1374 and 1158) were much higher than in the reference ones. Since the absorption band at 1510 cm^−1^ generally does not undergo significant changes due to wood degradation [19,37,38], an increase in the calculated ratios can be taken as an indication of the depletion of polysaccharides. The selected carbohydrate bands are affected by both cellulose and hemicelluloses, except for the band at 897 cm^−1^, which is specific for cellulose. Thus, the fact that the I_1510/897_ ratio in all the archaeological samples was higher than in the reference sample, indicates that the chemical changes involved not only hemicelluloses but also cellulose. The highest ratios were obtained for archaeological samples P1 (pine) and S1 (spruce) for both considered bands at 1374 cm^−1^ and 897 cm^−1^, which were the most degraded samples studied from a polysaccharide point of view.

### 2.2. Py-GC/MS

To assess the state of preservation of the archaeological wooden samples described in Section 4.1, analytical pyrolysis coupled with gas, chromatography and mass spectrometry (Py-GC/MS) was used. Figure 3 reports two chromatographic profiles obtained for archaeological samples S2 and P2 in spruce and pine wood, respectively. Only quantitative differences were observed, resulting in the same pyrolysis products. Pyrolysis profiles of the archaeological wood samples showed the same pyrolysis products as for the reference, but in different relative amounts. Since all the samples were analysed under the same instrumental conditions, the different relative abundances obtained are the result of the changes present in the material at the molecular level, which influenced the relative yields of the pyrolytic processes. Both species of archaeological wood analysed belong to softwood; therefore, their lignin is also mainly composed of guaiacyl units.

The degree of preservation of the archaeological wood was evaluated by comparing the results obtained with those of fresh, undegraded wood of the same species. The ratio between holocellulose and lignin content is a commonly used parameter to evaluate the degradation state of archaeological wood in terms of loss of one wood component (holocellulose or lignin) with respect to the other one, usually the loss of polysaccharides; therefore, a decrease in H/L is observed when compared with sound undegraded wood [39].

Table 2 shows the pyrolytic composition of the wood fractions for all the archaeological samples compared to those of undegraded reference wood. All the archaeological samples analysed have similar percentage values of the fractions to those of the reference samples. This can indicate no loss of the polysaccharide fraction with respect to the lignin, an effect contrary to that often found in archaeological wood [21,22,40], or both holocellulose and lignin are involved in the degradation at comparable level. However, the simple estimation of the H/L ratio is not sufficient in some cases to correctly reflect the chemical variations in the preservation of archaeological wood samples [39].

For a complete picture, to obtain further information on the degradation of the individual components of the wood, holocellulose and lignin, were analysed separately. Figure 4 shows the profiles obtained by categorising the pyrolysis products of both holocellulose (Figure 4a) and lignin (Figure 4b) according to their molecular structure and pyrolytic information [41,42] (Table 5).

As expected, undegraded pine and spruce woods showed a similar distribution of holocellulose pyrolysis products (Figure 4a), comparable abundances were obtained. Analysis of the categorised holocellulose products in the archaeological samples indicated an altered state of the polysaccharides. All archaeological wood samples exhibited very different profiles from those of reference samples (Figure 4a). A significant increase in the abundance of anhydrosugars and a decrease in cyclopentenones in all archaeological samples was observed. Among anhydrosugars, levoglucosan is the most abundant and it is reported in the literature that the pyrolysis yield of levoglucosan increases as the polymerisation degree of cellulose decreases [21,43]. Consequently, an increase in anhydrosugars can be considered as an indication of a partial depolymerisation of polysaccharides, a decay phenomenon that cannot be evaluated on the basis of the H/L ratio. This indicates a decrease in the degree of polymerisation of the polysaccharide matrix.

Differences in lignin profiles between the reference and archaeological wood samples were observed. Monomers are the pyrolysis products of lignin that are most abundantly formed in non-degraded, reference woods. Decrease of monomers with relative increase of other products with an altered side chain (short and long side chain), indicates an alteration/fragmentation of lignin propanoid side chains, Figure 4b. The depolymerisation effect was observed for all the pine and spruce archaeological wood samples. An increase in products with carbonyl and carboxyl functionalities in the samples collected from P1, P2 and S1 compared to the reference samples may be due to the mild oxidation of the lignin polymer.

### 2.3. ED-XRF and SEM-EDS

A non-destructive spectral investigation of wood samples (described in Section 4.1) confirmed their contamination with mineral substances. In some samples, ED-XRF analysis revealed a significant increase in the amount of iron signals compared to the reference (control) sample. This result was obtained for four out of five of the studied samples (S1, P1, P2, and P3), for which the level of Fe impulses counted ranged from 2 to even 20 times compared to the control sample (Figure 5a,b). Sample S2 had a comparable iron level with the reference (control) sample, but higher counts of nickel and lead (Figure 5c). The difference in impulses counted was about 3 or 4 times for lead and 10 times for nickel in samples S2 and P1. In a few samples, a high number of zinc impulses was also observed (Figure 5d).

Confirmation of the presence of mineral deposits in the analysed wood was provided by SEM-EDS studies. Due to the simultaneous observation of the surface at a significant magnification and the spectral analysis, data indicating the presence of various minerals was obtained (Figure 6). For sample S2, silver and sulphur signals were noted, which could indicate silver (I) sulfide, because the atomic content of silver (71% Ag) and sulphur (29% S) is stoichiometrically appropriate for the formation of Ag_2_S (Figure 6a). In addition, for sample P1, information on iron, arsenic and sulphur signals was collected, which may be associated with the presence of arsenopyrite (Figure 6c). In the same sample, in a different microarea, signals of lead and sulphur were recorded, indicating the presence of PbS (Figure 6b). There were numerous iron peaks in the spectra of samples S1, P1, P2 and P3, accompanied by sulphur signals. This may indicate the presence of iron sulphides, including pyrite or pyrrhotite (Figure 6d).

### 2.4. XRD

Table 3 reports the main crystalline phases, identified through X-ray diffraction analysis. The pyrite (FeS_2_, isometric crystal system) was the main phase identified, as also suggested by the high amount of iron and sulfur recorded in the microchemical analyses. Greigite (Fe^2 +^ Fe^3 +^ _2_S_4_, isometric crystal system) was identified in the samples collected from samples S2, P1, P2 and P3, with magnetite (Fe^2 +^ Fe^3 +^ _2_O_4,_ isometric crystal system) in samples P2 and P3. Greigite is the sulfur analog of magnetite and has a similar inverse spinel structure, but is metastable with respect to pyrite and marcasite [44].

It was not possible to calculate the crystallinity index of decayed wood with the XRD data because the crystalline part of the cellulose had totally disappeared (Figure 7).

## 3. Discussion

The unusual conditions in which this wooden structure worked make it an interesting object to study. The simultaneous influence of the highly destructive factors for wood, such as water and mechanical load as well as inorganic substances, also translate into interesting research results.

The results obtained by FT-IR and Py-GC/MS for the organic fraction of these wooden artefacts confirmed a very altered conservation condition in all the fragments studied. The FT-IR data showed a strong depletion of the polysaccharide component, which was confirmed by Py-GC/MS, through in-depth analysis of categorised pyrolysis products. Ratios between the relative intensities of lignin against the carbohydrate absorption bands from FTIR spectra highlighted that both cellulose and hemicelluloses had undergone decomposition.

The pyrolysis results underlined similar percentage values of the wood fractions to those of the reference samples, which is unusual for archaeological waterlogged wood, as found in a previous study performed on archaeological wood with a high content of inorganic salts from a salt mine [9]. In this case the H/L ratio was not useful enough to highlight degradation phenomena. Only in-depth analyses of categorised pyrolysis products indicated a strong depletion of polysaccharides. In wet archaeological wood, the degradation of the polysaccharide component is accompanied by a loss of the material due to the circulation of water, which dissolves the smallest molecules and eliminates them from the system. In this case, the degraded products of hemicelluloses and cellulose continued to be trapped in the complex wood matrix, resulting in unchanged content of the wood fractions. The lignin also underwent depolymerisation, and in those samples that showed a greater depletion of polysaccharides, this was accompanied by oxidation.

The results of the ED-XRF and SEM-EDS confirm that the wood samples analysed had been contaminated by the mineral substances. The level of the contamination varied. For some samples, the amount of iron shown as the number of the impulses counted was as much as 10–20 times greater than in the control sample. Some samples also showed a comparable level of iron, but with a much higher content of nickel or lead compared to the control sample. From the SEM-EDS studies carried out with a significant magnification, the presence of various mineral substances—quite typical for Lower Silesia—was confirmed, for example arsenopyrite, pyrrhotite and pyrite, deposits of which are found in Złoty Stok and the surrounding area [45,46,47,48].

XRD measurements were used to identify many iron- and sulphur-based compounds, which are key elements in wood degradation [49,50]. Pyrite (FeS_2_) is also frequently reported with mackinawite (FeS), greigite (Fe_3_S_4_), pyrrhotite (Fe1−xS), marcasite (orthorhombic polymorph of FeS_2_) and elemental sulphur (α-S8) [51]. The complexity of the stability of iron sulfides has been described in numerous studies. In particular, the formation of greigite may be related to the exposure to air of mackinawite. In this case, greigite is an intermediate transient compound to elemental sulphur and ferric oxyhydroxides (γ-FeOOH lepidocrocite, α-FeOOH goethite) and/or Fe_3_O_4_ magnetite. On the other hand, mackinawite can be oxidised into pyrite, with greigite as an intermediate compound under anoxic and acidic conditions, with a supply of sulphur via hydrogen sulphide production (or a loss of iron). Pyrite can be also formed; however, an excess of sulphur is required (Fe/S ratio smaller than 3/4). This excess may come from particular bacterial activity that leads to unusual acidity and/or a high H_2_Saq concentration. O_2_ and H_2_S concentrations, temperature, humidity and pH are the main parameters that explain the stability of these iron sulphides [52,53].

The environmental conditions of this gold mine, such as temperature, humidity, and the high concentration of inorganic compounds led to strong degradation of the lignocellulosic matrix. The presence of inorganic salts together with the degree of degradation of the wooden artefacts should be considered in the selection of the appropriate stabilisation method. According to the literature, the presence of some consolidating materials used for wood, such as PEG [50,54], in conjunction with iron salts can considerably worsen the conditions of wooden artefacts. In this case, treatment based on lactitol, trehalose or other non-hydrolysing sugars is recommended [55,56].

## 4. Materials and Methods

### 4.1. Sampling

The research material consisted of samples from the paternoster pump from the gold mine in Złoty Stok. The wood from Złoty Stok dates mainly to the 17th and 18th centuries, but wood from the 20th century was also found [57]. The studied elements were buried in a collapsed ruined gold mine and were completely submerged in water. The samples selected for investigation dated to the beginning of the 16th century. The parts of pump included the troughs that drain pumped water from the mine (and thus in constant contact with water) and a ladder (Figure 8).

All the elements found were selected for conservation, which was a limitation. During the sampling, efforts were made to minimise the interference with the objects so as not to affect their aesthetics. The samples came from the surface layers or from places damaged during the exploration of the objects and had a wet weight ranging from 0.7 to 5 g.

Samples S1, S2 and P1 were taken from the troughs and P2 and P3 from the ladder samples. Microscopic investigations showed that S1 and S2 were made of Norway spruce (*Picea abies* L.), while the others were made of Scots pine (*Pinus sylvestris* L.).

The weights of the samples in the state of maximum saturation with water and in the absolutely dry state were used to calculate the maximum water content (MWC) and basic density (BD) [58]. The ash content was determined on the basis of the TAPPI T 211 om-02 standard [59]. The results of BD and percentage content of ash are presented in Table 4.

### 4.2. FTIR

Before the measurements, all samples were ground and dried in an oven for 24 h at 40 °C and palletized with potassium bromide (KBr). Samples spectra were recorded as 32 scans at 4 cm^−1^ spectral resolution in the range of 400–4000 cm^−1^, using an FTIR IRPrestige 21 spectrometer (Shimadzu Corp., Tokyo, Japan). Scans were accumulated in transmission mode and analysed with Spectragryph v. 1.2.16 (Spectroscopy Ninja, Oberstdorf, Germany).

An already published methodology was used to calculate the lignin to carbohydrate ratio [19,35]. Peak areas were measured by integrating the area under the peak between two points relative to a baseline. Peak heights were also measured. The calculation was based on ratios of area or height of peaks specific for lignin 1510 cm^−1^ and carbohydrate groups 1374, 1158 and 897 cm^−1^.

### 4.3. ED-XRF and SEM-EDS

Energy dispersive X-ray fluorescence spectrometry (ED-XRF) and scanning electron microscopy coupled with an energy dispersion spectrometer (SEM-EDS), were used to study the wood samples. The analyses were performed on the XRF Spectro Midex X-ray fluorescence spectrometer with a molybdenum X-ray tube and excitation energy of 44.6 keV. The spectrometer is equipped with a Peltier cooled SDD detector. During the XRF studies, number of impulses of each element in the wood was compared to the control sample. SEM microscopic observations and EDS elemental analysis were performed on a Hitachi TableTop TM4000 table-top with an Oxford Instruments spectrometer. To the study a BSE mode, an accelerating voltage of 15 kV and charge-up reduction mode were used. The samples were observed at various magnifications from 25× to 1000×.

### 4.4. XRD

Archaeological wood samples were accompanied by fresh (non-degraded) wood samples for comparison. The samples belonged to two wood species, spruce (*Picea abies*) and pine (*Pinus sylvestris*). The sampled elements also showed strong contamination from metal salts. The evaluation of the crystalline component and in particular the presence of crystalline phases in the wood samples named S1, S2, P1, P2 and P3 were investigated by X-ray diffraction.

An X’Pert Pro PANalytical diffractometer equipped with an X’Celerator detector with a Cu X-ray tube (λ = 1.54 Å) was used with a Ni-filtered Cu–Kα radiation source. The X-ray tube was operated at 40 kV and 30 mA. The diffraction patterns were collected under the following conditions: 2θ range 3–70°, step size 0.02° and total time per pattern of 16 min 27 s. Soller and anti-scatter slits were used on the incident and diffracted beams; a divergent slit was used on the incident beam. A zero background sample stage was used. The phase identification of the samples was performed using the X’Pert HighScore program and the ICCD database.

### 4.5. Py-GC/MS

Pyrolysis was performed at 550 °C for 0.2 min using a micro-furnace of a multi-shot pyrolyzer EGA/Py-3030D (Frontier Lab) connected to GC 6890 Agilent Technologies (USA) and mass selective detector Agilent 5973. The formation of pyrolysis products occurred in the presence of 2 µL of derivatising agent 1,1,1,3,3,3-hexamethyldisilazane (HMDS, chemical purity 99.9%, Sigma Aldrich, Inc., St. Louis, MO, USA) for the thermally assisted silylation of pyrolysis products. Other operating conditions are detailed in [62].

Products were identified by comparing their mass spectra with spectra reported in the Wiley and NIST08 libraries and in the literature [21,40] and reported in Table 5 Deconvolution and integration of selected peaks formed from lignin and holocellulose were carried out by AMDIS software (Automated Mass spectral Deconvolution & Identification System by NIST). Semi-quantitative calculations were performed using normalised chromatographic areas, and the data were averaged and expressed as percentages. Lignin and holocellulose pyrolysis products were sorted into six and five groups, respectively (Table 5) as follows: lignin: monomers, long side chain, short side chain, demethylated, carbonyl and acid; and holocellulose: furans, cyclopentenones, pyranones, hydroxybenzenes and anhydrosugars.

## Figures and Tables

**Figure 1 molecules-27-05228-f001:**
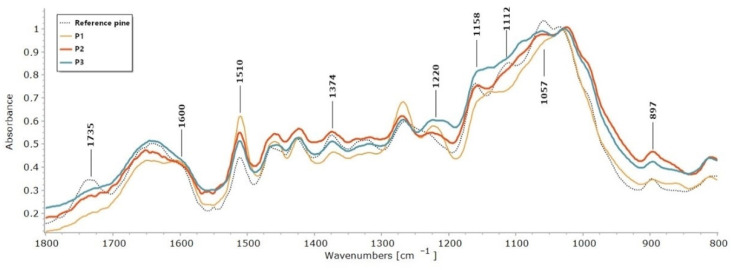
FTIR spectra of archaeological pine wood samples and reference pine wood sample.

**Figure 2 molecules-27-05228-f002:**
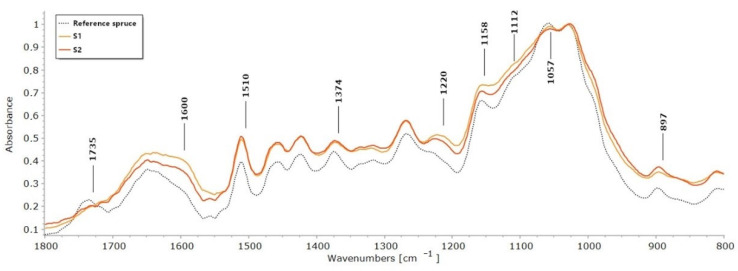
FTIR spectra of archaeological spruce wood samples and reference spruce wood sample.

**Figure 3 molecules-27-05228-f003:**
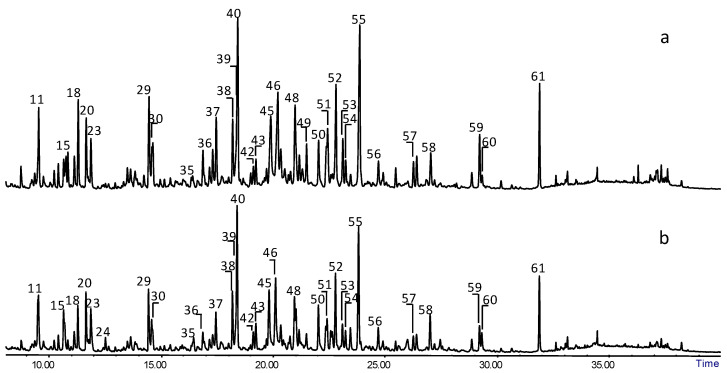
Py-GC-MS chromatographic profiles obtained for archaeological wood sample (**a**) S2 (spruce) and (**b**) P2 (pine). Numbers refer to Table 5.

**Figure 4 molecules-27-05228-f004:**
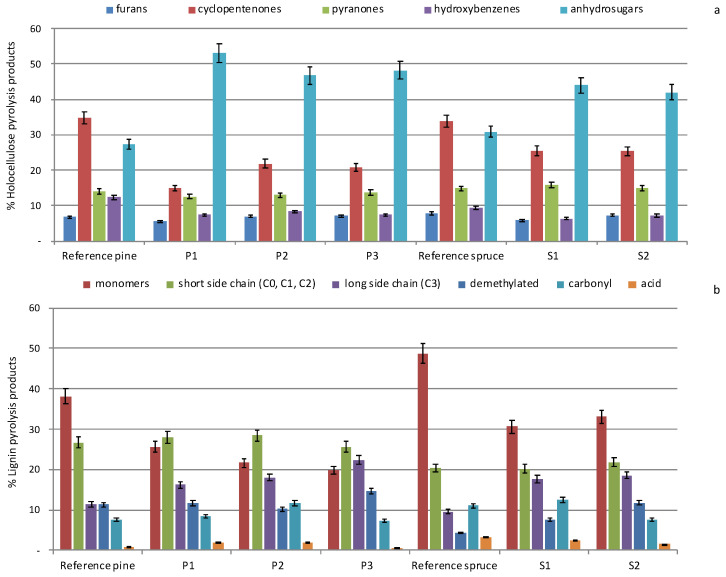
Distribution of (**a**) holocellulose and (**b**) lignin pyrolysis products from sound pine (Reference pine) and spruce (Reference spruce) wood samples, and archaeological pine (P1, P2 and P3) and spruce (S1 and S2) wood specimens. Relative abundances are expressed as percentages relative to total holocellulose (**a**) and total lignin fractions (**b**), respectively.

**Figure 5 molecules-27-05228-f005:**
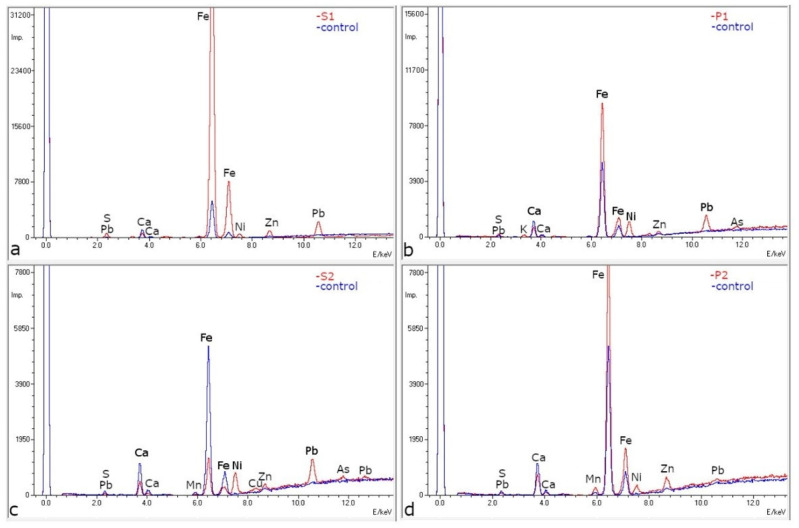
Selected ED-XRF spectra of archaeological wood samples S1 (**a**), P1 (**b**), S2 (**c**) and P2 (**d**) and reference wood samples (control) for P1 and P2 pine reference wood and for S1 and S2 spruce reference wood.

**Figure 6 molecules-27-05228-f006:**
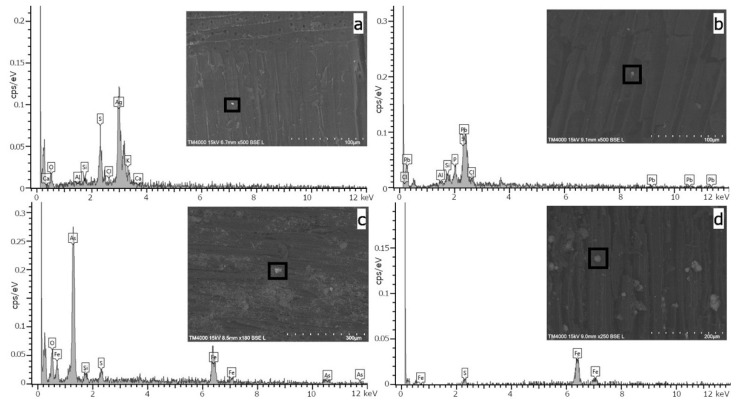
SEM images and EDS spectra for selected archaeological samples: S2 (**a**), P1 (**b**,**c**) and P3 (**d**).

**Figure 7 molecules-27-05228-f007:**
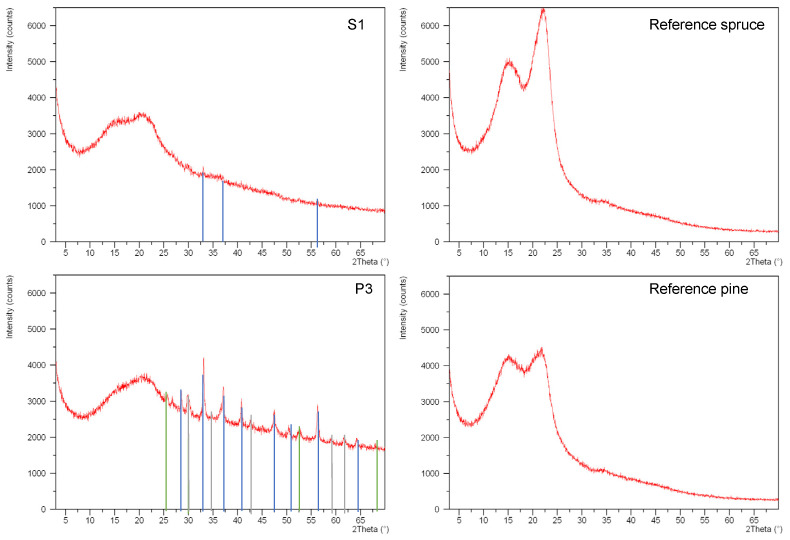
X-ray patterns of samples S1 and P3 with related spruce and pine references (in blue lines diffraction peaks of pyrite, in green greigite, in gray magnetite).

**Figure 8 molecules-27-05228-f008:**
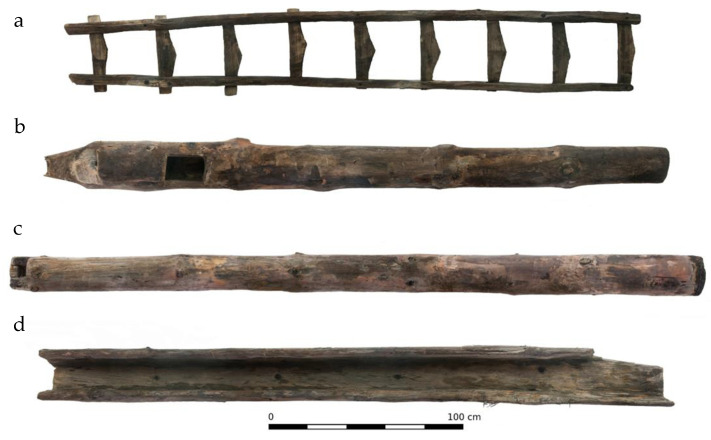
Wooden elements from a paternoster pump, where (**a**) ladder and the mine drainage system: (**b**,**c**) pipes, and (**d**) trough.

**Table 1 molecules-27-05228-t001:** Intensity of lignin (L, band at 1510 cm^−1^)/carbohydrate ratios (bands at 1374 cm^−1^ and 1158 cm^−1^—cellulose and hemicelluloses; band at 897 cm^−1^—cellulose) obtained for archaeological and reference wood samples, described in Section 4.1.

	Reference Pine	P1	P2	P3	Reference Spruce	S1	S2
High I_L_/I_1374_	2.04	8.21	3.95	3.39	2.31	3.88	4.09
Area I_L_/I_1374_	2.14	8.55	3.66	3.34	2.09	3.62	3.71
High I_L_/I_1158_	1.07	–	1.82	–	1.17	1.77	1.75
Area I_L_/I_1158_	1.11	–	1.59	–	0.99	1.46	1.47
High I_L_/I_897_	3.61	20.48	5.06	6.39	4.05	8.06	5.32
Area I_L_/I_897_	4.24	26.10	5.80	8.58	4.41	9.30	5.95

**Table 2 molecules-27-05228-t002:** Wood composition determined by Py-GC-MS, where H-holocellulose, L-lignin and H/L ratio.

Wood Fraction	Reference Pine	P1	P2	P3	Reference Spruce	S1	S2
H [%]	73.6	65.2	73.7	80.3	70.5	74.8	79.0
L [%]	26.4	34.8	26.3	19.7	29.5	25.2	21.0
H/L	2.8 ± 0.3	1.9 ± 0.1	2.9 ± 0.4	4.1 ± 0.3	2.4 ± 0.2	3.0 ± 0.2	3.8 ± 0.3

**Table 3 molecules-27-05228-t003:** Main crystalline phases identified via Xray diffraction.

Archaeological Samples	Crystalline Phase	Formula
S1	Pyrite	FeS_2_
S2	Greigite	Fe_2_S_4_
P1	Pyrite, greigite	FeS_2_, Fe_2_S_4_
P2	Pyrite, greigite, magnetite	FeS_2_, Fe_2_S_4_, Fe_3_O_4_
P3	Pyrite, greigite, magnetite	FeS_2_, Fe_2_S_4_, Fe_3_O_4_

**Table 4 molecules-27-05228-t004:** BD and percentage ash content in the collected samples.

	Reference Pine	P1	P2	P3	Reference Spruce	S1	S2
BD (kg/m^3^)	418 *	213	385	351	403 *	221	359
Ash (%)	0.2–0.5 **	1.70	1.54	5.85	0.2–0.5 **	12.01	2.41

* Basic density of fresh wood [60]. ** The ash content of fresh woods from temperate zones [61].

**Table 5 molecules-27-05228-t005:** The pyrolysis products identified by Py-GC/MS and categorised were H-holocellulose, L-lignin, lignin units: p-hydroxyphenyl (H-lignin), and guaiacyl (G-lignin), in bold more abundant fragments.

	Compound	m/z	Category	Origin
1	1,2-dihydroxyethane (2TMS)	73, 103, **147**, 191	small molecules	H/L
2	2-hydroxymethylfuran (TMS)	53, 73, **81**, 111, 125, 142, 155, 170	furan	H
3	phenol (TMS)	75, **151**, 166	short side chain	H-lignin
4	2-hydroxypropanoic acid (2TMS)	73, 117, **147**, 190	small molecules	H/L
5	2-hydroxyacetic acid (2TMS)	73, **147**, 177, 205	small molecules	H/L
6	1-hydroxy-1-cyclopenten-3-one (TMS)	53, 73, 81, 101, 111, 127, **155**, 169	cyclopentenone	H
7	3-hydroxymethylfuran (TMS)	53, 75, **81**, 111, 125, 142, 155, 170	furan	H
8	o-cresol (TMS)	73, 91, 135, 149, **165**, 180	short side chain	H-lignin
9	2-furancarboxylic acid (TMS)	73, 95, **125**, 169, 184	furan	H
10	m-cresol (TMS)	73, 91, **165**, 180	short side chain	H-lignin
11	2-hydroxy-1-cyclopenten-3-one (TMS)	53, 73, 81, 101, 111, 127, **155**, 170	cyclopentenone	H
12	p-cresol (TMS)	73, 91, **165**, 180	short side chain	H-lignin
13	3-hydroxy-(2H)-pyran-2-one (TMS)	75, 95, 125, 151, **169**, 184	pyran	H
14	Z-2,3-dihydroxy-cyclopent-2-enone (TMS)	59, **73**, 115, 143, 171, 186	cyclopentenone	H
15	*E*-2,3-dihydroxy-cyclopent-2-enone (TMS)	75, 101, **143**, 171, 186	cyclopentenone	H
16	1,2-dihydroxybenzene (TMS)	**75**, 91, 136, 151, 167, 182	hydroxybenzene	H/L
17	3-hydroxy-(4H)-pyran-4-one (TMS)	75, 95, 139, 151, **169**, 184	pyran	H
18	5-hydroxy-2H-pyran-4(3H)-one (TMS)	59, 75, 101, 129, 143, **171**, 186	pyran	H
19	2-hydroxymethyl-3-methy-2-cyclopentenone (TMS)	**73**, 103, 129, 173, 183, 198	cyclopentenone	H
20	1-hydroxy-2-methyl-1-cyclopenten-3-one (TMS)	73, 97, 125, 139, **169**, 184	cyclopentenone	H
21	1-methy-2-hydroxy-1-cyclopenten-3-one (TMS)	73, 97, 125, 139, **169**, 184	cyclopentenone	H
22	1,3-dihydroxyacetone (2TMS)	**73**, 103, 147, 189, 219	small molecules	H/L
23	guaiacol (TMS)	73, 151, **166**, 181, 196	short side chain	G-lignin
24	3-hydroxy-6-methyl-(2H)-pyran-2-one (TMS)	73, 109, 139, 168, **183**, 198	pyran	H
25	2-methyl-3-hydroxy-(4H)-pyran-4-one (TMS)	73, 101, 153, **183**, 198	pyran	H
26	2-methyl-3-hydroxymethyl-2-cyclopentenone (TMS)	**73**, 103, 129, 173, 183, 198	cyclopentenone	H
27	2,3-dihydrofuran-2,3-diol (2TMS)	73, 147, **231**, 246	furan	H
28	2-furyl-hydroxymethylketone (TMS)	73, 81, 103, 125, **183**, 198	furan	H
29	5-hydroxymethyl-2-furaldehyde (TMS)	73, 81, 109, 111, 139, 169, **183**, 198	furan	H
30	4-methylguaiacol (TMS)	73, 149, **180**, 195, 210	short side chain	G-lignin
31	1,2-dihydroxybenzene (2TMS)	**73**, 151, 239, 254	hydroxybenzene	H/L
32	2-hydroxymethyl-2,3-dihydropyran-4-one (TMS)	73, 142, 170, 185, 200	pyran	H
33	1,4:3,6-dianhydro-α-D-glucopyranose (TMS)	**73**, 103, 129, 155, 170, 171, 186	anhydrosugars	H
34	Z-2,3-dihydroxy-cyclopent-2-enone (2TMS)	73, 147, 230, **243**, 258	cyclopentenone	H
35	4-methylcatechol (2TMS)	**73**,180, 253, 268	demethylated	G-lignin
36	4-ethylguaiacol (TMS)	73, 149, 179, **194**, 209, 224	short side chain	G-lignin
37	1,4-dihydroxybenzene (2TMS)	73, 112, **239**, 354	hydroxybenzene	H/L
38	4-vinylguaiacol (TMS)	73, 162, 177, **192**, 207, 222	short side chain	G-lignin
39	3-hydroxy-2-hydroxymethyl-2-cyclopentenone (2TMS)	73, 147, **257**, 272	cyclopentenone	H
40	E-2,3-dihydroxy-cyclopent-2-enone (2TMS)	73, 147, **243**, 258	cyclopentenone	H
41	4-ethylcatechol (2TMS)	**73**, 147, 179, 231, 267, 282	demethylated	G-lignin
42	3-hydroxy-2-(hydroxymethyl) cyclopenta-2,4-dienone (2TMS)	73, 147, **255**, 270	cyclopentenone	H
43	eugenol (TMS)	73, 147, 179, **206**, 221, 236	long side chain	G-lignin
44	3-methoxy-1,2-benzenediol (2TMS)	**73**, 153, 254, 269, 284	demethylated	G-lignin
45	3,5-dihydroxy-2-methyl-(4H)-pyran-4-one (2TMS)	73, 128, 147, 183, **271**, 286	pyran	H
46	Z-isoeugenol (TMS)	73, 179, **206**, 221, 236	long side chain	G-lignin
47	vanillyl alcohol (2TMS)	**73**, 151, 210, 253, 268, 283, 298	long side chain	G-lignin
48	vanillin (TMS)	73, **194**, 209, 224	carbonyl	G-lignin
49	1,2,3-trihydroxybenzene (3TMS)	73, 133, 147, **239**, 327, 342	hydroxybenzene	H
50	*E*-isoeugenol (TMS)	73, 179, **206**, 221, 236	long side chain	G-lignin
51	1,2,4-trihydroxybenzene (3TMS)	73, 133, 147, 239, 327, **342**	hydroxybenzene	H
52	acetovanillone (TMS)	73, **193**, 208, 223, 238	carbonyl	G-lignin
53	1,4-anydro-D-galactopyranose (3TMS)	**73**, 129, 147, 157, 191, 204, 217, 243, 332	anhydrosugars	H
54	2,3,5-trihydroxy-4H-pyran-4-one (3TMS)	**73**, 133, 147, 239, 255, 270, 330, 345, 360	pyran	H
55	1,6-anydro-beta-D-glucopyranose (3TMS)	**73**, 103, 129, 147, 191, 204, 217, 243, 333	anhydrosugars	H
56	1,4-anhydro-D-glucopyranose (3TMS)	**73**, 103, 129, 147, 191, 204, 217, 243, 332	anhydrosugars	H
57	1,6-anydro-beta-D-glucofuranose (3TMS)	**73**, 103, 129, 147, 191, 204, 217, 243, 319	anhydrosugars	H
58	vanillic acid (2TMS)	73, 253, 282, **297**, 312	acid	G-lignin
59	*Z*-coniferyl alcohol (2TMS)	73, 204, 252, 293, 309, **324**	monomer	G-lignin
60	coniferylaldehyde (TMS)	73, 192, 220, 235, 250	carbonyl	G-lignin
61	*E*-coniferyl alcohol (2 TMS)	73, 204, 252, 293, 309, **324**	monomer	G-lignin

## Data Availability

Not applicable.

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
