# Peer review of "Gold Mine Wooden Artefacts: Multianalytical Investigations for the Selection of Appropriate Consolidation Treatments"

_molecules, 2022, doi:10.3390/molecules27165228_

Round 1

Reviewer 1 Report

This paper aims to show the condition of wooden materials with multi-analytical approach and to select the appropriate conservation treatment. The manuscript is quite well written and will be interesting for Molecules journal readers. I have some recommendations to improve the text.

In my opinion, the first part of the aim was fulfilled very well, on the other hand, only little attention was paid to conservation (lines 305-310) and that only based on published information, not based on performed experiments. Perhaps the authors should consider modifying the aim of this manuscript.

Please change “hemicellulose” to “hemicelluloses” in the whole manuscript, there is not only one hemicellulose in wood, but several types of hemicelluloses.

Lines 184-186. Authors state: All the samples analyzed, the polysaccharides show a good state of preservation compared with undegraded wood. The lignin fraction showed the greatest degradation. Elsewhere in the manuscript they state: The FT-IR data showed a strong depletion of the polysaccharide component, which was confirmed by Py-GC/MS. Ratios between the relative intensities of lignin against the carbohydrate absorption bands from FTIR spectra highlighted that both cellulose and hemicellulose had undergone decomposition (lines 263-266). Please, address this more clearly.

I understand that samples from No. 1 were labeled in the experiments, but the working descriptions (6-11) should not be used for publication. In addition, it would be useful to publish photos of the samples used.

Author Response

Review 1

This paper aims to show the condition of wooden materials with multi-analytical approach and to select the appropriate conservation treatment. The manuscript is quite well written and will be interesting for Molecules journal readers. I have some recommendations to improve the text.

Thanks for your comments and overall concern with our assumptions in relation to the analysis we have conducted on the archaeological waterlogged material.

1) In my opinion, the first part of the aim was fulfilled very well, on the other hand, only little attention was paid to conservation (lines 305-310) and that only based on published information, not based on performed experiments. Perhaps the authors should consider modifying the aim of this manuscript.

Response: The authors modified the purpose of the study to emphasize the importance of research on the degree of wood degradation and the presence of elements in the selection of the preservation method.

In the introduction lines 103-110:

In this study, the chemical transformation of lignocellulosic polymers and contamination with inorganic salts in the archaeological wood from the Złoty Stok gold mine were investigated using complementary instrumental techniques: FT-IR, Py-GC/MS, ED-XRF, SEM-EDS 104 and XRD. Due to the unique burial place of the studied objects, it was expected that they would have unusual chemical properties, necessitating the use of special conservation procedures. The aim of the study was to identify the chemical properties of wood, which will be helpful in choosing a preservation method that gives a long-lasting effect of wood stabilization and protection against further degradation.

2) Please change “hemicellulose” to “hemicelluloses” in the whole manuscript, there is not only one hemicellulose in wood, but several types of hemicelluloses.

Response: The authors made the necessary corrections throughout the text

3) Lines 184-186. Authors state: All the samples analyzed, the polysaccharides show a good state of preservation compared with undegraded wood. The lignin fraction showed the greatest degradation. Elsewhere in the manuscript they state: The FT-IR data showed a strong depletion of the polysaccharide component, which was confirmed by Py-GC/MS. Ratios between the relative intensities of lignin against the carbohydrate absorption bands from FTIR spectra highlighted that both cellulose and hemicellulose had undergone decomposition (lines 263-266). Please, address this more clearly.

Response: As suggested by the Reviewer, the text has been supplemented to clarify the aspects related to the interpretation of the FTIR and Py-GC/MS analysis results

4) I understand that samples from No. 1 were labeled in the experiments, but the working descriptions (6-11) should not be used for publication. In addition, it would be useful to publish photos of the samples used.

Response: As suggested by the Reviewer, new numbering of samples and photos of the elements were added.

Reviewer 2 Report

Review of the article: Gold Mine Wooden Artefacts: Multianalytical Investigations for the Selection of Appropriate Consolidation Treatments

The topic of this research and output might be useful for the Molecules readers. The conducted work is worth to publish, however needs many corrections. In order to improve the manuscript the following suggestions should be considered.   Introduction §  Line 38: …very wet or very dry - these are relative terms. Please specify the thresholds for wet and dry wood protection condition. §  Lines 41 - 44: The high temperatures, variable and generally high humidity, variable oxygen content, numerous metal-wood contact points, and the high content of inorganic compounds have a significant impact on the biotic and abiotic degradation factors. This is a repetition of an abstract sentence (Lines 16 - 19). Please change the form of the sentence to avoid repetition. §  Lines 56 - 59: The number should be placed after the author's name i.e.: …Wang et al. [10]… §  Lines 68 - 70: Why does one sentence form one paragraph? §  Line 72: …ATR… First, give the full name of the method, test, etc., and then the abbreviation in brackets. §  Line 83: …hemicellulose… Should be hemicelluloses. Please improve throughout the article.   §  Why are the results discussed first and then the Research material presented? Please correct the order of the chapters. §  Please add a section - Conclusion.   §  Line 115: However, in all the woods studied… General record - woods. In all tested wood species? It should be indicated what kind of wood in terms of species. §  Line 121: …hemicellulose-related bands… Please correct …hemicelluloses-related bands…   Figure 4: §  % Holocellulosepyrolysis products Or Holocellulose pyrolysis products (%) §  Please correct Figure 4b. The Y axes should have the same value - this makes it easier to compare the data (Figure 4a and 4b). In addition, the values in Figure 4b are outside the scale of the Y axis. Figure 5: §  Why are some peaks off the Y axis? Figure 7: §  The Figures are presented in an illegible way. Data on the axes are unreadable.   §  Why The Discussion is ahead of the description - Materials and Methods?   Discussion §  Lines 302 - 304: Why does one sentence form one paragraph? §  In the title was written: …for the Selection of Appropriate Consolidation Treatments. Why is this part only a few sentences dedicated to this? – Lines 305 – 310.   Materials and Methods §  Line 337: …KBR… See comments for Line 72.  

Author Response

Review 2

Review of the article: Gold Mine Wooden Artefacts: Multianalytical Investigations for the Selection of Appropriate Consolidation Treatments

The topic of this research and output might be useful for the Molecules readers. The conducted work is worth to publish, however needs many corrections. In order to improve the manuscript the following suggestions should be considered.  

Introduction § 

Line 38: …very wet or very dry - these are relative terms. Please specify the thresholds for wet and dry wood protection condition.

Response: Sentence has been reworded and references added (lines 36-39).

  • Lines 41 - 44: The high temperatures, variable and generally high humidity, variable oxygen content, numerous metal-wood contact points, and the high content of inorganic compounds have a significant impact on the biotic and abiotic degradation factors. This is a repetition of an abstract sentence (Lines 16 - 19). Please change the form of the sentence to avoid repetition.

Response: we disagree, the repetition of the specific conditions found in the mines, in the introduction and in the abstract makes sense. We find no reason to change these conditions which are listed. it is possible to change the order of the list, but it does not change the sense.

  • Lines 56 - 59: The number should be placed after the author's name i.e.: …Wang et al. [10]…

Response: was corrected

  • Lines 68 - 70: Why does one sentence form one paragraph?

Response: has been merged into the next paragraph

  • Line 72: …ATR… First, give the full name of the method, test, etc., and then the abbreviation in brackets.

Response: was corrected

  • Line 83: …hemicellulose… Should be hemicelluloses. Please improve throughout the article.  

Response: We have checked and corrected

  • Why are the results discussed first and then the Research material presented? Please correct the order of the chapters.

Response: unfortunately, this objection was also made by the other reviewers, we agree with you but the Molecules journal provides this order of paragraphs. The structure of the publication was prepared according to the author's guide. However, references to subsequent chapters describing the material studied have been added.

  • Please add a section - Conclusion.

Response: Dear Reviewer for the Molecules journal the paragraph of "Conclusion" is not mandatory, for this reason we have not included, since the concluding arguments have been made in the “Discussion” paragraph and it would be only a repetition.

  • Line 115: However, in all the woods studied… General record - woods. In all tested wood species? It should be indicated what kind of wood in terms of species.

Response: the sentence has been rephrased (lines 119-120)

  • Line 121: …hemicellulose-related bands… Please correct …hemicelluloses-related bands…

Response: We have checked and corrected

Figure 4: §  % Holocellulosepyrolysis products Or Holocellulose pyrolysis products (%) §  Please correct Figure 4b. The Y axes should have the same value - this makes it easier to compare the data (Figure 4a and 4b). In addition, the values in Figure 4b are outside the scale of the Y axis.

Response: the changes to Figure 4 have been applied

Figure 5: §  Why are some peaks off the Y axis?

Response: Because there are weaker peaks of the determined and labelled elements, like Pb, the Figure 5 was modified, and all peaks are signed.

Figure 7: §  The Figures are presented in an illegible way. Data on the axes are unreadable.

Response: the changes to Figure 7 have been applied

  • Why The Discussion is ahead of the description - Materials and Methods? Discussion

Response: unfortunately, this objection was also made by the other reviewers, we agree with you, but the Molecules journal provides this order of paragraphs.

The structure of the publication was prepared according to the author's guide. However, references to subsequent chapters describing the material studied have been added

  • Lines 302 - 304: Why does one sentence form one paragraph?

Response: has been merged into the next paragraph

  • In the title was written: …for the Selection of Appropriate Consolidation Treatments. Why is this part only a few sentences dedicated to this? – Lines 305 – 310.  

Response: In our article, we have described the analytical techniques that can be used to obtain important information about the wood, which is important for selection a method of its conservation. Without this information, in our case about the degree of carbohydrate and lignin degradation as well as the content of mineral substances, there is a great risk of choosing the wrong, e.g. the most popular method of conservation, which, due to its ineffectiveness, may contribute to the further degradation of historic wood.

In order to emphasize the importance of the research described for the choice of the conservation method, the purpose of the study was modified.

Materials and Methods

  • Line 337: …KBR… See comments for Line 72.

Response: was corrected

Reviewer 3 Report

On reading the title and introduction to this paper, I assumed I would be reading an analytical study of waterlogged wood from a mine site that led to the design of a conservation protocol. This was not the case. There is a highly detailed analytical study but the one sentence at the end of the discussion corresponding to treatment options does not justify the title. However, I liked the organisation of the study and it is detailed and sound, and worth publishing, but with a more indicative title, and after revision of the text.

Overall, this is a very nice study but the presentation is at times confusing for the reader. The information is there but not always so clear and this stems from three things (1) the materials and methods are after the results so it was not clear what I was even going to read about in the results before starting and then I had to keep referring to the end of the text for clarification, (2) there are no images of the wood, and (3) there are some apparently but not actually contradictory results from the FTIR and py-GCMS and no acknowledgement is made of this fact or that it is explained later in the discussion. Before publication, I think the title and the results section text need a rework to make sure the points are clear and the materials needs to go before the results. 

The following are my specific comments on the text:

Line 23, use commas instead of hyphens

line 26: perhaps you can preface this with the fact that these are all iron containing minerals, as many readers will be aware of the potential issues with humidity, wood and iron

L38: It must be noted that when the conditions are very wet, they must also be free of oxygen

L39: Could you give a brief summary of the specific conditions, this sentence is rather vague

L47: please reword, so as not to use various twice in the same sentence

L108: It is very unusual to go from the introduction to the results without any explanation of the materials and methods. I found myself confused at reading the detailed FTIR analysis of object 8, 9 and 11, with no explanation of these objects previously so I cannot tell if the FTIR data is expected for an object of its condition or not. Please rework this and put the materials and methods before the results section so that the reader can clearly understand the data presented. For example the spectra for object 8 is different to 9 and 11 although they are the same wood. Is this something to do with the condition? It would be very helpful to include a picture of the wood before the FTIR is presented (after the methods and materials have been described) to make the data interpretation clearer.

L161: This is very interesting, again it would be useful to see the samples to appreciate this data

L172: No comma between gas and chromatography

L174: This statement is a little confusing, is the point you are making that the products are the same but the quantities are different? This could be reworded for clarity

L181-186: The pyrolysis-GCMS seems to be in direct contradiction to the FTIR data, namely the py-GCMS shows that the lignin is the most degraded and the FTIR says it is the celluloses on the same artefacts. How do you explain this? Further into this section around L200-211, you mention that both the lignin and the holocellulose are degraded. It is rather difficult to follow the data interpretation as it appears to be often contradictory. 

L225: please keep the notation standard, sometimes object 7 sometimes sample 7, please use one or the other for clarity throughout

L263-271: In this section the FTIR and py-GCMS are now compared. Although it is clear already from the previous results section that there is an unusual contradiction between the FTIR and py-GCMS the reason is not given until line 271. It would be helpful for the reader if the authors acknowledge already in the results section that there is an apparent mismatch and explain the reason is discussed below. I must admit I found the presentation of these results confusing until I got to this line, where it all made sense. It would help the impact of the paper to acknowledge this earlier on.

L271: Is this because they are only wet from the environment but not submerged?

L305: For the lack of in-depth discussion, or justification at all for the consolidation treatment it is not worth to include this paragraph. It appears to only exist to justify the title, which in my opinion should be changed as it is not indicative of the content of the paper

Author Response

Review 3

On reading the title and introduction to this paper, I assumed I would be reading an analytical study of waterlogged wood from a mine site that led to the design of a conservation protocol. This was not the case. There is a highly detailed analytical study but the one sentence at the end of the discussion corresponding to treatment options does not justify the title. However, I liked the organisation of the study and it is detailed and sound, and worth publishing, but with a more indicative title, and after revision of the text.

Response: The title of the article contains, in accordance with its content, information on the selection of the preservation method based on the results of chemical analysis. The title does not contain information on the preservation process, which is quite well described in the literature and is a reproductive action in accordance with the guidelines for the concentration of the impregnant and the frequency of its change. In our article, we have described the analytical techniques that can be used to obtain important information about the wood, which is important for selection a method of its conservation. Without this information, in our case about the degree of carbohydrate and lignin degradation as well as the content of mineral substances, there is a great risk of choosing the wrong, e.g. the most popular method of conservation, which, due to its ineffectiveness, may contribute to the further degradation of historic wood.

In order to emphasize the importance of the research described for the choice of the conservation method, the purpose of the study was modified.

Overall, this is a very nice study but the presentation is at times confusing for the reader. The information is there but not always so clear and this stems from three things

(1) the materials and methods are after the results so it was not clear what I was even going to read about in the results before starting and then I had to keep referring to the end of the text for clarification,

Response: unfortunately, this objection was also made by the other reviewers, we agree with you, but the Molecules journal provides this order of paragraphs,

The structure of the publication was prepared according to the author's guide. However, references to subsequent chapters describing the material studied have been added.

(2) there are no images of the wood, and

Response: Photos of the elements were included in the publication.

(3) there are some apparently but not actually contradictory results from the FTIR and py-GCMS and no acknowledgement is made of this fact or that it is explained later in the discussion. Before publication, I think the title and the results section text need a rework to make sure the points are clear and the materials needs to go before the results. 

Response: In order to clarify the aspects of the interpretation of the results of FTIR and Py-GC/MS analysis, subsection 2.5 Py-GC/MS has been significantly modified. It was explained in detail which information about wood degradation is collected thanks to the FTIR analysis and which by Py-GC/MS.

The following are my specific comments on the text:

Line 23, use commas instead of hyphens

Response: we disagree, hyphens are correct form and not commas when interfacing techniques (combined) are used.

line 26: perhaps you can preface this with the fact that these are all iron containing minerals, as many readers will be aware of the potential issues with humidity, wood and iron

Response: information that minerals are iron based was added in the sentence (line 26)

L38: It must be noted that when the conditions are very wet, they must also be free of oxygen

Response: The sentence: “It is one of the most resistant organic materials, however wooden objects can be preserved for long periods of time only under particular conditions, which need to be very wet or very dry”.

Was change for:

It is one of the most resistant organic materials, however wooden objects can be preserved for long periods of time only under particular conditions. In a temperate climate, these are the anaerobic conditions of wet environments and the aerobic conditions of dry environments, considering the low intensity of wood degradation factors. (lines 37-40)

L39: Could you give a brief summary of the specific conditions, this sentence is rather vague

Response: The specific conditions mentioned in sentence 39 are explained on lines 41-42:

“The high temperatures, variable and L41 generally high humidity, variable oxygen content, numerous metal-wood contact points, L42 and the high content of inorganic compounds…”

L47: please reword, so as not to use various twice in the same sentence

Response: The sentence: “In the literature there are various studies carried out on wood from various types of mines.”

was change in:

In the literature there are many studies carried out on wood from various types of mines. (line 48)

L108: It is very unusual to go from the introduction to the results without any explanation of the materials and methods.

I found myself confused at reading the detailed FTIR analysis of object 8, 9 and 11, with no explanation of these objects previously so I cannot tell if the FTIR data is expected for an object of its condition or not. Please rework this and put the materials and methods before the results section so that the reader can clearly understand the data presented. For example the spectra for object 8 is different to 9 and 11 although they are the same wood. Is this something to do with the condition? It would be very helpful to include a picture of the wood before the FTIR is presented (after the methods and materials have been described) to make the data interpretation clearer.

Response: Unfortunately, this objection was also made by the other reviewers, we agree with you, but the Molecules journal provides this order of paragraphs. The structure of the publication was prepared according to the author's guide. However, references to subsequent chapters describing the material studied have been added.

The picture was also added to the text (Figure 8).

L161: This is very interesting, again it would be useful to see the samples to appreciate this data

Response: The photos of the elements were added.

L172: No comma between gas and chromatography

L174: This statement is a little confusing, is the point you are making that the products are the same but the quantities are different? This could be reworded for clarity

Response: In order to clarify the aspects of the interpretation of the results of FTIR and Py-GC/MS analysis, subsection 2.5 Py-GC/MS. has been significantly modified. It was explained in detail which information about wood degradation is collected thanks to the FTIR analysis and which thanks to Py-GC/MS.

L181-186: The pyrolysis-GCMS seems to be in direct contradiction to the FTIR data, namely the py-GCMS shows that the lignin is the most degraded and the FTIR says it is the celluloses on the same artefacts. How do you explain this? Further into this section around L200-211, you mention that both the lignin and the holocellulose are degraded. It is rather difficult to follow the data interpretation as it appears to be often contradictory. 

Response: In order to clarify the aspects of the interpretation of the results of FTIR and Py-GC/MS analysis, subsection 2.5 Py-GC/MS has been significantly modified. It was explained in detail which information about wood degradation is collected thanks to the FTIR analysis and which by Py-GC/MS.

L225: please keep the notation standard, sometimes object 7 sometimes sample 7, please use one or the other for clarity throughout

Response: The naming/making of samples has been unified

L263-271: In this section the FTIR and py-GCMS are now compared. Although it is clear already from the previous results section that there is an unusual contradiction between the FTIR and py-GCMS the reason is not given until line 271. It would be helpful for the reader if the authors acknowledge already in the results section that there is an apparent mismatch and explain the reason is discussed below. I must admit I found the presentation of these results confusing until I got to this line, where it all made sense. It would help the impact of the paper to acknowledge this earlier on.

Response: In order to clarify the aspects of the interpretation of the results of FTIR and Py-GC/MS analysis, subsection 2.5 Py-GC/MS has been significantly modified. It was explained in detail which information about wood degradation is collected thanks to the FTIR analysis and which by Py-GC/MS.

L271: Is this because they are only wet from the environment but not submerged?

Response: “The studied elements were buried in a collapsed ruined gold mine and were completely submerged in water.” The sentence in red was added to the text lines 350-351.

L305: For the lack of in-depth discussion, or justification at all for the consolidation treatment it is not worth to include this paragraph. It appears to only exist to justify the title, which in my opinion should be changed as it is not indicative of the content of the paper

Response: As was explained at the beginning of answer to review the title of the article contains, in accordance with its content, information on the selection of the preservation method based on the results of chemical analysis. The title does not contain information on the preservation process, which is quite well described in the literature and is a reproductive action in accordance with the guidelines for the concentration of the impregnant and the frequency of its change. In our article, we have described the analytical techniques that can be used to obtain important information about the wood, which is important for selection a method of its conservation. Without this information, in our case about the degree of carbohydrate and lignin degradation as well as the content of mineral substances, there is a great risk of choosing the wrong, e.g. the most popular method of conservation, which, due to its ineffectiveness, may contribute to the degradation of historic wood.

In order to emphasize the importance of the research described for the choice of the conservation method, the purpose of the study was modified

Round 2

Reviewer 1 Report

In my opinion, all reviewers' questions and suggestions were accepted or explained, thus this manuscript can be published in its present form.

Reviewer 2 Report

I accept the changes made.

In the case of Figure 8, add the reference/letters a, b, c, d and give them also in the caption Figure 8 (Line 363). Thanks to this, the description will be more accurate and will point to the specific tested/described item.